# Soluble tumor necrosis factor receptor 2 is associated with progressive diabetic kidney disease in patients with type 2 diabetes mellitus

Tsung-Hui Wu[1,2], Li-Hsin Chang[3,4], Chia-Huei Chu[5,6], Chii-Min Hwu[1,2], Harn-Shen Chen[1,2], Liang-Yu Lin[1,2]*

1 Division of Endocrinology and Metabolism, Department of Medicine, Taipei Veterans General Hospital, Taipei, Taiwan, 2 Faculty of Medicine, National Yang Ming Chiao Tung University, Taipei, Taiwan, 3 Division of Endocrinology and Metabolism, Department of Medicine, Yeezen General Hospital, Taoyuan, Taiwan, 4 Department of Medical Laboratory Science and Biotechnology, Yuanpei University of Medical Technology, Hsinchu, Taiwan, 5 Department of Otorhinolaryngology-Head and Neck Surgery, Mackay Memorial Hospital, Taipei, Taiwan, 6 Department of Audiology and Speech Language Pathology, Mackay Medical College, New Taipei City, Taiwan

* linly@vghtpe.gov.tw

**Data Availability Statement:** All relevant data are within the paper and its Supporting information files.

## Abstract

### Background

Chronic low-grade inflammation is considered one of the major mechanisms for the progression of diabetic kidney disease. We investigated the prognostic value of circulating soluble tumor necrosis factor receptor 2 (sTNFR2) for early nephropathy in patients with type 2 diabetes.

### Materials and methods

A total of 364 patients with type 2 diabetes and an estimated glomerular filtration rate (eGFR) $\geq$30 mL/min/1.73m$^2$ were followed up for a median of 4 years. Renal outcomes were defined as a composite of either or both a >30% decline in the eGFR and/or albuminuria stage progression determined with consecutive tests.

### Results

Seventy-three patients developed renal composite events. Serum concentrations of sTNFR2 were strongly associated with the risk of renal function decline and progressive changes in albuminuria. Through a receiver operating characteristic curve analysis, a serum sTNFR2 level of 1.608 ng/mL was adopted as the discriminator value for predicting renal outcomes (area under the curve 0.63, 95% confidence interval 0.57–0.70, p < 0.001), yielding a sensitivity of 75.3% and a specificity of 51.2%. The association of sTNFR2 levels $\geq$1.608 ng/mL to renal outcomes was significant after adjusting for relevant variables (hazard ratio 2.27, 95% confidence interval 1.23–4.20, p = 0.009) and remained consistent

**Funding:** This study was supported by research grants V104E11-004-MY2, V105C-131, V107C-201, V108C-197, V109C-179, V110C-198, and V111D62-002-MY3-1 provided to L.Y.L. from Taipei Veterans General Hospital and No. 2021001 to L.H.C. from Yeezen General Hospital. The funders had no role in study design, data collection and analysis, decision to publish, or preparation of the manuscript.

**Competing interests:** The authors have declared that no competing interests exist.

across subgroups stratified by age, sex, systolic blood pressure, eGFR, albuminuria, and the use of renin-angiotensin system blockers.

## Conclusions

Higher circulating levels of sTNFR2 are independently associated with an eGFR decline and progressive albuminuria in patients with type 2 diabetes.

## Introduction

Although there have been improvements in diabetes management, the prevalence of diabetic kidney disease (DKD) continues to rise due to improved patient prognoses and the growing incidence of diabetes mellitus [1]. Approximately 35%-50% of patients with type 2 diabetes will eventually develop various kidney disorders, such as microalbuminuria, macroalbuminuria, or impaired renal function [1, 2]. Diabetic kidney disease is the most common cause of end stage renal disease (ESRD) worldwide [3], and the health care costs of DKD has increased substantially in the recent decades [2]. Chronic kidney disease is the dominant contributor to excess mortality in patients with type 2 diabetes [4]. To prevent ESRD and premature mortality, it is crucial to identify patients with high risk of advanced renal disease at an early stage and provide personalized management.

Screening of patients with diabetic nephropathy includes measuring their urinary albumin-creatinine ratio (UACR) and estimated glomerular filtration rate (eGFR) at least once annually [5]. Although most patients with a risk of DKD can be identified by screening for albuminuria and eGFR, significant glomerular structural changes are often already developed by the time microalbuminuria becomes apparent [6]. In addition, patients with type 2 diabetes can develop renal impairments in the absence of albuminuria [7]; thus making albuminuria an ineffective screening test for these patients. Novel biomarkers are needed to detect the early stages of diabetic kidney disease.

Activation of the tumor necrosis factor-α pathway has emerged as an important mechanism in the pathogenesis of DKD [8]. Tumor necrosis factor-α is an inflammatory cytokine involved in a number of signaling pathways that lead to apoptosis and inflammatory processes, by interacting with two types of membrane receptors: tumor necrosis factor receptor (TNFR) 1 and TNFR2 [9]. The circulating level of soluble tumor necrosis factor receptor 2 (sTNFR2), which is released from the cell surface, is associated with end-stage renal disease, cardiovascular disease, and mortality in patients with type 2 diabetes [10–13]. However, the association of sTNFR2 with the development of early nephropathy in patients with type 2 diabetes has been investigated less often. Therefore, the aim of this study was to explore the association of sTNFR2 with early changes in diabetic nephropathy in patients with type 2 diabetes.

## Materials and methods

### Study design and participants

Patients with type 2 diabetes mellitus, visiting the endocrinology and metabolism outpatient clinic at Taipei Veterans General Hospital between June 2014 and December 2019, were enrolled in the prospective observational cohort study. Patients with an eGFR <30 mL/min/1.73m$^2$, as determined by the Modification of Diet in Renal Disease equation, or elevated serum levels of alanine aminotransferase greater than two-fold the upper normal limit,

uncontrolled cancer, or hemoglobin A1C (HbA1C) ≥9.0% were excluded. Baseline clinical parameters were registered, including age, sex, height, weight, blood pressure, and hip and waist circumferences. The following risk factors and comorbidities were evaluated: cardiovascular disease, hyperlipidemia, hypertension, retinopathy, neuropathy, and smoking status. Retinopathy was assessed by means of nonmydriatic fundus photography. Participants were diagnosed as having neuropathy by neurologists or by electrodiagnostic testing. Serum creatinine, HbA1C, lipid profiles, and urine albumin and creatinine levels were measured every 6 months in the central laboratory of the Department of Clinical Pathology and Laboratory of Taipei Veterans General Hospital. The urinary albumin-creatinine ratio was used as a measure of albuminuria. Normoalbuminuria, microalbuminuria, and macroalbuminuria were defined as a UACR <30 mg/g creatinine, 30–299 mg/g creatinine, and ≥300 mg/g creatinine, respectively. We measured serum sTNFR2 levels according to the manufacturer's instructions (Quantikine ELISA kit; R&D Systems, Minneapolis, MN, USA). The intra-assay coefficient of variation was 2.6–4.8%, while the inter-assay coefficient of variation was 3.5–5.1%. The participants were followed until they developed renal composite events or until December 2019.

This prospective study was approved by the ethics committee at Taipei Veterans General Hospital. Written informed consent was obtained from each participant before entering the study. The study complied with the guidelines of the Declaration of Helsinki.

## Renal outcomes

The renal outcomes were a composite of sustained decreases in the eGFR and/or the progression of albuminuria stage. The sustained decreases in the eGFR were defined as a >30% reduction in the eGFR from baseline in at least two consecutive samples. The progression of albuminuria stages was defined as the transition from normoalbuminuria to microalbuminuria or macroalbuminuria, or from microalbuminuria to macroalbuminuria. We confirmed albuminuria progression if patients developed transition to a more advanced stage of albuminuria from baseline in two consecutive follow-up at 6-month intervals.

## Statistical analyses

The variables were expressed as percentage values for categorical data and as mean ± standard deviation for continuous data. Univariate Cox proportional analyses were used to examine the relevant variables associated with renal composite outcomes. We examined the association between serum sTNFR2 levels and renal outcomes by multivariate Cox proportional hazard models that included covariates with $P < 0.05$ in the univariate Cox proportional analyses. The association between serum sTNFR2 levels and renal outcomes was expressed as hazard ratios (HRs) and 95% confidence intervals (95% CI). The following multivariate Cox models were built: model 1, adjusted for age and sex; model 2, adjusted for age, sex, and duration of diabetes; model 3, adjusted for retinopathy, neuropathy, systolic blood pressure, HbA1C, eGFR, UACR, RAS inhibitors, and diuretics in addition to all variables in model 2. A receiver operating characteristic curve analysis was performed, and the cut-off value of sTNFR2 for predicting renal composite events was determined on the basis of the maximum value of the Youden index. Differences between groups, which were stratified by the occurrence of renal outcomes or the cut-off value of sTNFR2, were analyzed by Pearson's chi-squared test for categorical data and Student's t-test for continuous variables. The survival curve for the renal composite events was illustrated using the Kaplan-Meier method, and different groups were compared using the log-rank test. A two-sided P value <0.05 was considered statistically significant. All analyses were performed with the use of SPSS Statistics version 22 (IBM Corporation, Armonk, NY, USA).

# Results

## Study patients

A total of 364 patients were enrolled in the study. The mean age of the patients was 61.4 years, and 70.3% of the patients were men. The mean duration of diabetes was 11.7 years. One-third of the subjects had albuminuria, and the prevalence of retinopathy and neuropathy was 9.3% and 9.1%. Sixty-seven (18.4%) patients had stage 3 chronic kidney disease. The mean HbA1C was 7.1%, mean blood pressure was 114.0 mmHg, and mean body mass index (BMI) was 26.3 kg/m$^2$. The baseline characteristics of the study population are summarized in Table 1.

**Table 1. Baseline characteristics of the entire study population and according to the occurrence of the renal outcomes.**

| | Entire cohort | Occurrence of the renal outcomes | | P value |
|---|---|---|---|---|
| | (n = 364) | Yes (n = 73) | No (n = 291) | |
| Age | 61.4±12.8 | 63.6±11.6 | 60.9±13.0 | 0.102 |
| Male sex (%) | 256 (70.3) | 47 (64.4) | 209 (71.8) | 0.214 |
| Smoking (%) | 111 (30.5) | 18 (24.7) | 93 (32.0) | 0.226 |
| Coronary artery disease (%) | 72 (19.8) | 19 (26.0) | 53 (18.2) | 0.134 |
| Hyperlipidemia (%) | 295 (81.0) | 61 (83.6) | 234 (80.4) | 0.539 |
| Hypertension (%) | 224 (61.5) | 49 (67.1) | 175 (60.1) | 0.273 |
| Retinopathy (%) | 34 (9.3) | 15 (20.5) | 19 (6.5) | <0.001 |
| Neuropathy (%) | 33 (9.1) | 12 (16.4) | 21 (7.2) | 0.014 |
| Albuminuria (%) | 121 (33.2) | 36 (49.3) | 85 (29.2) | 0.001 |
| CKD stage 3 (%) | 67 (18.4) | 21 (28.8) | 46 (15.8) | 0.011 |
| CKD stage 3 with albuminuria (%) | 41 (11.3) | 16 (21.9) | 25 (8.6) | 0.001 |
| Duration of diabetes (year) | 11.7±7.5 | 14.7±9.1 | 10.9±6.9 | 0.001 |
| Body mass index | 26.3±4.4 | 26.4±5.6 | 26.3±4.1 | 0.842 |
| Waist-hip ratio | 0.94±0.07 | 0.94±0.07 | 0.94±0.06 | 0.858 |
| Systolic blood pressure (mmHg) | 132.8±15.4 | 134.9±16.5 | 132.3±15.1 | 0.204 |
| Diastolic blood pressure (mmHg) | 77.3±10.7 | 78.4±12.0 | 77.0±10.3 | 0.299 |
| Mean blood pressure (mmHg) | 114.0±13.9 | 116.0±13.8 | 113.9±12.2 | 0.183 |
| Urinary albumin-creatinine ratio (mg/g Cr) | 104.8±301.9 | 258.1±521.9 | 64.7±191.7 | 0.003 |
| HbA1C (%) | 7.1±0.8 | 7.2±0.8 | 7.0±0.8 | 0.118 |
| Total cholesterol (mg/dL) | 166.8±28.1 | 167.1±26.0 | 166.7±28.7 | 0.914 |
| Creatinine (mg/dL) | 0.97±0.27 | 1.01±0.30 | 0.95±0.26 | 0.116 |
| eGFR (mL/min/1.73m$^2$) | 78.2±19.4 | 72.6±20.4 | 79.6±19.0 | 0.006 |
| sTNFR2 (ng/mL) | 2.00±1.17 | 2.35±1.15 | 1.92±1.15 | 0.004 |
| Metformin (%) | 283 (77.7) | 56 (76.7) | 227 (78.0) | 0.812 |
| Sulfonylurea (%) | 144 (39.6) | 32 (43.8) | 112 (38.5) | 0.403 |
| Dipeptidyl peptidase-4 inhibitor (%) | 68 (18.7) | 17 (23.3) | 51 (17.5) | 0.259 |
| Sodium/glucose co-transporter 2 inhibitor (%) | 13 (3.6) | 1 (1.4) | 12 (4.1) | 0.257 |
| Insulin (%) | 64 (17.6) | 17 (23.3) | 47 (16.2) | 0.152 |
| Renin-angiotensin system blockade (%) | 174 (47.8) | 45 (61.6) | 129 (44.3) | 0.008 |
| Diuretics (%) | 43 (11.8) | 15 (20.5) | 28 (9.6) | 0.010 |

Data are expressed as mean±SD for continuous variables and numbers and percentages for non-continuous variables. Differences between groups stratified by the occurrence of renal outcomes were analyzed by Student's t-test for continuous variables and Pearson's chi-squared test for non-continuous variables.

CKD, chronic kidney disease; Cr, creatinine; HbA1C, hemoglobin A1C; eGFR, estimated glomerular filtration rate; sTNFR2, soluble tumor necrosis factor receptor type 2.

## Association of sTNFR2 with the renal outcomes

Over a median follow-up time of 4 years, 73 participants had a renal composite event, which is composed of 46 events of progression of albuminuria stages and 27 events of a >30% reduction in the eGFR. Among the 46 events of progressive albuminuria, 27 participants had a transition from normoalbuminuria to microalbuminuria, and 18 participants progressed from microalbuminuria to macroalbuminuria. One patient had a rapid progression from normoalbuminuria to macroalbuminuria. Patients with renal composite events had longer durations of diabetes, higher percentages of retinopathy or neuropathy, higher UACRs, and lower eGFRs at baseline. The coverage of the renin-angiotensin system (RAS) blockade was more extensive in patients with renal outcomes (Table 1). Correlations between sTNFR2 concentrations and BMIs or waist-to-hip ratios were weak (Pearson's correlation coefficient -0.062 for sTNFR2 and BMI, p = 0.241; -0.021 for sTNFR2 and waist-hip-ratio, p = 0.686). The sTNFR2 concentration was associated with renal outcomes in the univariate Cox proportional analysis (hazard ratio [HR] 1.29, 95% confidence interval [CI] 1.09–1.52, p = 0.003). Other clinical covariates, such as the duration of diabetes, presence of retinopathy or neuropathy, eGFR and UACR at baseline, and the use of RAS inhibitors or diuretics were also associated with renal outcomes. The association of sTNFR2 concentration with renal events was attenuated after adjustment in sequential models (Table 2).

The ideal cut-off point for sTNFR2 concentration is the point where true-positive findings and few false-positive results are detected in most patients. The optimal cut-off value of sTNFR2 levels obtained using the receiver operating characteristic curve analysis was 1.608 ng/mL, allowing for a sensitivity of 75.3% and a specificity of 51.2%. The area under the

**Table 2. Results of the univariate and multivariate Cox proportional hazard models for the association of sTNFR2 with renal composite events.**

|  | HR | 95% CI | P value |
|---|---|---|---|
| sTNFR2 | 1.29 | 1.09–1.52 | 0.003 |
| sTNFR2 ≥1.608 ng/mL | 2.47 | 1.45–4.21 | 0.001 |
| Duration of diabetes | 1.05 | 1.02–1.08 | <0.001 |
| Retinopathy | 2.78 | 1.58–4.92 | <0.001 |
| Neuropathy | 1.86 | 1.00–3.46 | 0.049 |
| eGFR | 0.98 | 0.97–1.00 | 0.009 |
| UACR (per 100 mg/g Cr increase) | 1.10 | 1.06–1.14 | <0.001 |
| RAS inhibitors | 1.73 | 1.08–2.77 | 0.023 |
| Diuretics | 1.78 | 1.01–3.15 | 0.046 |
| Model 1 for sTNFR2 | 1.27 | 1.07–1.52 | 0.007 |
| Model 2 for sTNFR2 | 1.24 | 1.03–1.50 | 0.023 |
| Model 3 for sTNFR2 | 1.11 | 0.88–1.41 | 0.394 |
| Model 1 for sTNFR2 ≥1.608 ng/mL | 2.41 | 1.38–4.21 | 0.002 |
| Model 2 for sTNFR2 ≥1.608 ng/mL | 2.31 | 1.32–4.04 | 0.003 |
| Model 3 for sTNFR2 ≥1.608 ng/mL | 2.27 | 1.23–4.20 | 0.009 |

Model 1: adjusted for age and sex.

Model 2: adjusted for age, sex, and duration of diabetes.

Model 3: adjusted for age, sex, duration of diabetes, retinopathy, neuropathy, systolic blood pressure, HbA1C, eGFR, UACR, RAS inhibitors, and diuretics.

HR, hazard ratio; CI, confidence interval; sTNFR2, soluble tumor necrosis factor receptor type 2; eGFR, estimated glomerular filtration rate; UACR, urinary albumin-creatinine ratio; Cr, creatinine; RAS, renin-angiotensin system; HbA1C, hemoglobin A1C.

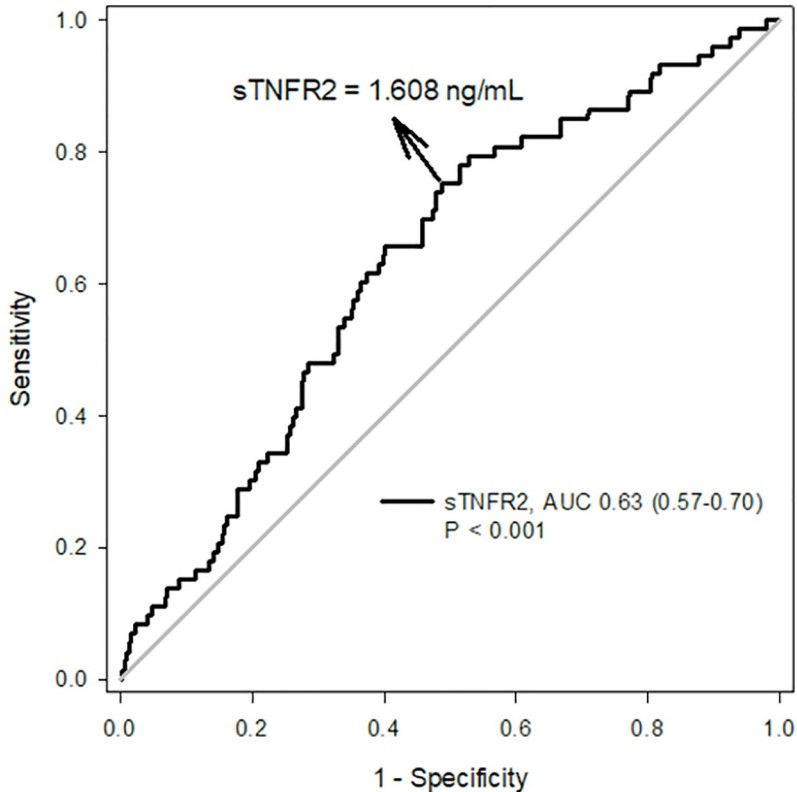

**Fig 1. Receiver operating characteristic curves of soluble tumor necrosis factor receptor 2 (sTNFR2) in patients with type 2 diabetes.**

receiver operating characteristic curve of sTNFR2 was 0.63 (95% CI 0.57–0.70, p < 0.001; Fig 1). The highest tertile of sTNFR2 concentration had an increased risk of renal composite events (p = 0.012 from the log-rank test; Fig 2a).

### Association of sTNFR2 levels ≥1.608 ng/mL with the renal outcomes

The clinical characteristics of the study population stratified based on sTNFR2 concentrations (≥1.608 ng/mL or <1.608 ng/mL) are shown in S1 Table. The patients with sTNFR2 concentrations ≥1.608 ng/mL were older and had longer durations of diabetes, higher percentage of retinopathy, and higher UACRs and lower eGFRs at baseline. The coverage of the RAS blockade was more extensive in patients with sTNFR2 concentrations ≥1.608 ng/mL than in patients with sTNFR2 concentrations <1.608 ng/mL (55.3% vs. 38.9%, p = 0.002). The patients with sTNFR2 concentrations ≥1.608 ng/mL had more renal composite events at the end of the study than those with sTNFR2 concentrations <1.608 ng/mL did (27.9% vs. 10.8%; p < 0.001 from the log-rank test; Fig 2b). Regarding the separate components of the renal composite events, the frequency of each was also higher in the study group with sTNFR2 concentrations ≥1.608 ng/mL (13.7% vs 6.6% for eGFR decline >30%, p = 0.027; 18.3% vs. 6.6% for worsening albuminuria, p = 0.001; S1 Table).

In the univariate Cox proportional analysis, sTNFR2 concentrations ≥1.608 ng/mL were associated with higher renal event rates (HR 2.47, 95% CI 1.45–4.21, p = 0.001). A sTNFR2 concentration ≥1.608 ng/mL predicted renal events in all of the multivariate models (HR 2.41,

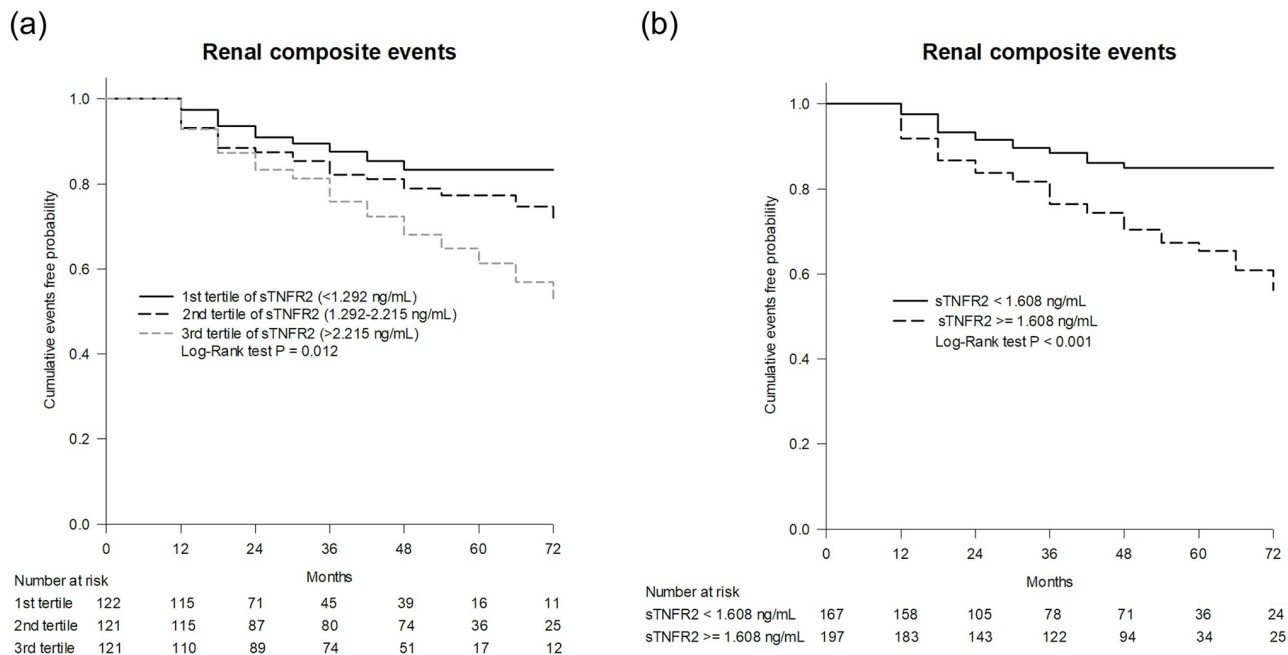

**Fig 2. Kaplan-Meier curves of cumulative event-free survival of renal events (a composite of a >30% reduction of the estimated glomerular filtration rate and/or the progression of albuminuria stages) in patients with type 2 diabetes according to: (a) the tertiles of soluble tumor necrosis factor receptor 2 (sTNFR2) levels and (b) the threshold value of sTNFR2 (≥1.608 ng/mL vs. <1.608 ng/mL).** Differences between curves were analyzed with log-rank statistics.

95% CI 1.38–4.21, p = 0.002 for model 1; HR 2.31, 95% CI 1.32–4.04, p = 0.003 for model 2; HR 2.27, 95% CI 1.23–4.20, p = 0.009 for model 3; Fig 3).

We performed a subgroup analysis based on age (younger or older than 60 years), sex, systolic blood pressure (≥ or <140 mmHg), eGFR (≥ or <60 mL/min/1.73m²), UACR (≥ or <30 mg/g creatinine), and the use of RAS inhibitors. The association of a sTNFR2 concentration ≥1.608 ng/mL with renal outcomes remained significant across all subgroups, including patients with an eGFR ≥60 mL/min/1.73m² or normoalbuminuria (Table 3). In the female participants, the association between sTNFR2 concentration ≥1.608 ng/mL and renal composite events was significant initially but became attenuated after adjustment in model 2 and model 3 (Crude HR 3.35, 95% CI 1.15–9.77, p = 0.027; HR 3.53, 95% CI 1.19–10.49, p = 0.023 for model 1; HR 3.00, 95% CI 0.98–9.15, p = 0.053 for model 2; HR 4.12, 95% CI 1.00–17.07, p = 0.051 for model 3).

## Discussion

In our study, elevated serum sTNFR2 concentrations were associated with worsening albuminuria and progressive eGFR decline in patients with type 2 diabetes. Circulating sTNFR2 levels ≥1.608 ng/mL were a strong predictor of the progression of albuminuria and renal impairment in patients with type 2 diabetes, and the association was independent of relevant clinical covariates, including age, duration of diabetes, baseline UACR and eGFR, and the use of RAS inhibitors. Comparing to the current screening strategies of diabetic nephropathy as measurement of UACR and eGFR, sTNFR2 is a biomarker to detect earlier stages of diabetic kidney disease. As a noninvasive and easily accessible method, measuring serum levels of sTNFR2 is a potential screening tool for diabetic nephropathy.

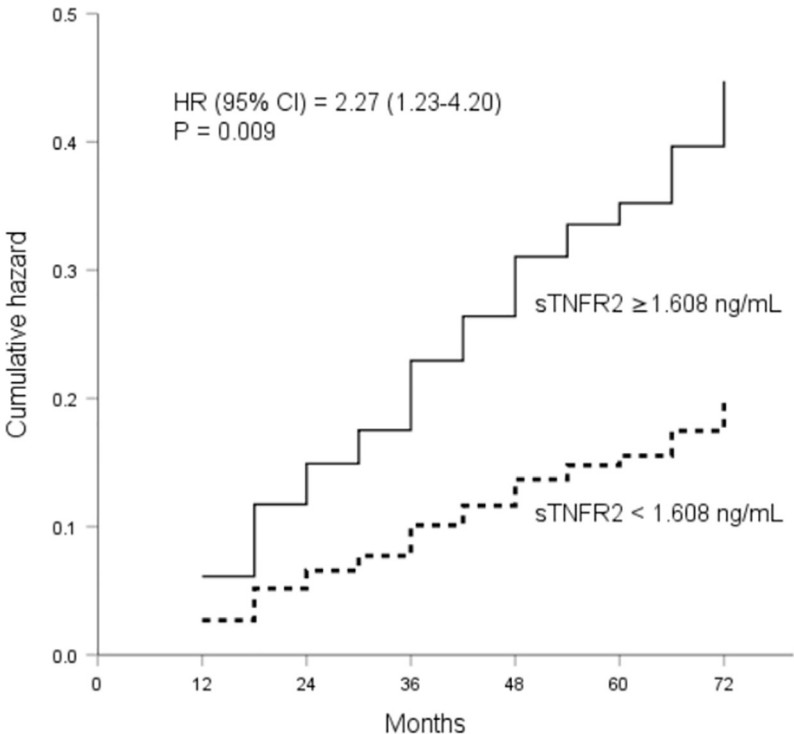

**Fig 3. Cumulative hazard curves for renal events (a composite of a >30% reduction of the estimated glomerular filtration rate and/or the progression of albuminuria stages) in patients with type 2 diabetes based on the threshold value of soluble tumor necrosis factor receptor 2 (≥1.608 ng/mL vs. <1.608 ng/mL).** Analysis results were adjusted for age, sex, duration of diabetes, retinopathy, neuropathy, systolic blood pressure, hemoglobin A1C, estimated glomerular filtration rate, urinary albumin-creatinine ratio, renin-angiotensin system inhibitors, and diuretics.

**Table 3. Results of the association between sTNFR2 concentrations ≥1.608 ng/mL and renal composite events stratified by age, sex, systolic blood pressure, eGFR, UACR, and the use of RAS inhibitors.**

| Subgroup | Crude HR | 95% CI | P value for interaction |
|---|---|---|---|
| Age ≥60 years old (n = 222) | 2.58 | 1.16–5.76 | 0.87 |
| Age <60 years old (n = 142) | 2.63 | 1.19–5.81 | |
| Male sex (n = 256) | 2.12 | 1.13–3.97 | 0.19 |
| Female sex (n = 108) | 3.35 | 1.15–9.77 | |
| SBP ≥140 mmHg (n = 110) | 3.49 | 1.19–10.26 | 0.62 |
| SBP <140 mmHg (n = 254) | 2.17 | 1.17–4.03 | |
| eGFR ≥60 ml/min/1.73m$^2$ (n = 297) | 2.64 | 1.45–4.82 | 0.39 |
| eGFR <60 ml/min/1.73m$^2$ (n = 67) | 0.84 | 0.25–2.85 | |
| UACR ≥30 mg/g Cr (n = 121) | 1.91 | 0.79–4.60 | 0.27 |
| UACR <30 mg/g Cr (n = 243) | 2.38 | 1.19–4.75 | |
| RAS inhibitors (n = 174) | 2.76 | 1.28–5.93 | 0.19 |
| No RAS inhibitors (n = 190) | 1.90 | 0.87–4.12 | |

HR, hazard ratio; CI, confidence interval; SBP, systolic blood pressure; eGFR, estimated glomerular filtration rate; UACR, urinary albumin-creatinine ratio; Cr, creatinine; RAS, renin-angiotensin system.

In addition to the traditional risk factors, such as poor glycemic [14] or blood pressure [15] control, chronic low-grade inflammation plays an important role in the development of DKD [8]. Activation of the tumor necrosis factor-α pathways is associated with renal function decline in patients with chronic kidney disease [16]. Tumor necrosis factor-α is involved in the signal transduction pathways by interacting with two types of membrane receptors: TNFR 1 and TNFR2 [17]. Tumor necrosis factor receptor 1 is mainly present in glomerular and tubular endothelial cells, whereas TNFR2 is usually absent in normal kidneys and transcriptionally expressed in the renal cells in various kidney diseases [18]. Tumor necrosis factor receptor 1 and TNFR2 induce shared and distinctive signalling pathways that lead to apoptosis, proliferation, and inflammation [8]. Both membrane-bound receptors TNFR1 and TNFR2 can be cleaved by the metalloproteinase and released as circulating polypeptides, soluble TNFR1 (sTNFR1) and sTNFR2 [19]. Higher serum sTNFR2 concentrations have been proven to be associated with increased risks of renal function decline or ESRD in patients with type 2 diabetes [10–13] and chronic kidney disease in patients with type 1 diabetes [20]. In one cohort of American Indians (Pima Indians) with type 2 diabetes, patients with macroalbuminuria whose serum sTNFR2 levels were in the highest quartile had a 88.7% cumulative incidence of ESRD, whereas those with macroalbuminuria and sTNFR2 levels in the lowest three quartiles had a 47.3% cumulative incidence of ESRD at 10 years of follow-up [12]. Higher sTNFR2 levels were associated with all-cause mortality as well [10]. Comparing to previous studies mainly focusing on advanced diabetic kidney disease, the findings in our cohort suggested that sTNFR2 is a promising biomarker in the early stages of diabetic nephropathy. The association of sTNFR2 concentration $\geq$1.608 ng/mL with renal outcomes remained significant in all subgroups, including patients with eGFRs $\geq$60 mL/min/1.73m$^2$ or normoalbuminuria.

The significance of the association between serum sTNFR2 levels and the renal outcomes disappear after adjusting several confounding factors in model 3 (Table 2). The failure to detect significant association may be due to inadequate sample size and duration of follow-up. Further research is needed to confirm the association of sTNFR2 with renal outcomes in diabetic patients.

Soluble tumor necrosis factor receptor 1 has been considered a valuable predictor of renal function decline among patients with type 2 diabetes [10–13, 21]. A recent meta-analysis demonstrated that both circulating sTNFR1 and sTNFR2 levels were independently associated with DKD progression in more than 5000 patients with diabetes [22]. We previously reported that serum sTNFR1 levels were associated with renal outcomes in patients with type 2 diabetes while 283 participants were enrolled in the same cohort in this study [9]. The association of serum sTNFR1 levels above the cut-off value yielded by the receiver operating characteristic curve analysis with DKD progression was significant after adjusting for relevant clinical variables (HR 2.43, 95% CI 1.18–5.02, p = 0.01). Although the hazard ratios calculated for renal outcomes according to the serum sTNFR1 levels were higher than those calculated for serum sTNFR2 levels, the CIs for the hazard ratios overlapped. In addition, both serum sTNFR1 and sTNFR2 levels were measured in the 364 patients enrolled in the current study. The difference between areas under the ROC curve of sTNFR1 and sTNFR2 is not significant (data not shown). Both sTNFR1 and sTNFR2 are potential biomarkers in the assessment of diabetic kidney disease.

Patients with type 2 diabetes and renal impairments do not always have preceding albuminuria [23]. In the United Kingdom Prospective Diabetes Study (UKPDS) cohort, 51% of the patients with renal impairments had normoalbuminuria for a median of 15 years of follow-up [24]. The presence of nonproteinuric renal dysfunction in patients with type 2 diabetes increases their risk of ESRD and cardiovascular disease [25]. Increased vascular resistance in renal interlobar arteries has been suggested to cause damage to nephron structures, contributing to nonproteinuric diabetic nephropathy [26]. Tumor necrosis factor-α is an essential

mediator of inflammation that can induce renal vasoconstriction, which is a potential mechanism of nonproteinuric diabetic nephropathy [27]. In our study, the association of serum sTNFR2 levels with renal composite events was consistent across multiple prespecified subgroups, including participants with and without albuminuria. This finding identified sTNFR2 as a potential predictor of renal function decline in patients with type 2 diabetes and normoalbuminuria.

There may be a racial disparity in serum sTNFR2 levels among patients with type 2 diabetes. The median serum sTNFR2 level in Pima Indians (4835 pg/mL) [12] was much higher than that in Caucasian patients in the Joslin Study of the Genetics of Type 2 Diabetes and Kidney Complications (2527 pg/mL) [11], although increased serum sTNFR2 levels in Pima Indians may be partly attributed to obesity. In a previous study comprising of Japanese patients [28], the median serum sTNFR2 level in participants with an eGFR $\geq$60 mL/min/1.73m$^2$ was 2972 pg/mL, which was comparable with that in Caucasians. The serum sTNFR2 levels in our study did not seem markedly different from those of previous studies, and the results from our cohort still imply that sTNFR2 could be a potential predictor of renal outcomes in patients with type 2 diabetes. Further studies are needed to clarify the normal ranges among different races.

The limitations of our study should be considered. First, the participants in this cohort were recruited from a veterans hospital which mainly serves male patients. It accounts for male-predominant cohort in this study. In the female participants, the borderline results of the association between sTNFR2 concentration $\geq$1.608 ng/mL and renal composite events may be due to relatively small sample size. Second, we estimated glomerular filtration rates from a serum creatinine-based equation, which are less accurate than estimates that are based on serum cystatin C or direct measurements of renal function. Direct measurement of the GFR may augment the strength of the association. Third, sodium/glucose co-transporter 2 (SGLT2) inhibitors and glucagon-like peptide-1 (GLP-1) analogues have renoprotective actions in patients with type 2 diabetes beyond glycemic control. Patients in our cohort rarely used sodium/glucose co-transporter 2 inhibitors or glucagon-like peptide-1 analogues because of the restricted reimbursement policies for them, owing to their high acquisition costs. Finally, the prospective observational design does not necessarily imply a causal relationship. Further studies are required to confirm that interventions to circulating sTNFR2 levels contribute to beneficial effects on DKD progression.

## Conclusions

Circulating sTNFR2 levels are associated with the progression of diabetic kidney disease in patients with type 2 diabetes independent of baseline albuminuria or renal function. Soluble tumor necrosis factor receptor 2 is a potential prognostic biomarker for the personalized management of worsening diabetic nephropathy.

## Supporting information

**S1 Table. Baseline characteristics and renal composite events of patients grouped according to soluble tumor necrosis factor receptor type 2 (sTNFR2) concentrations.**
(DOCX)

## Acknowledgments

The authors thank the Medical Sciences & Technology Building of Taipei Veterans General Hospital for providing experimental space and facilities. The authors declare no conflicts of interest regarding the contents of this manuscript.

## Author Contributions

**Conceptualization:** Tsung-Hui Wu, Li-Hsin Chang, Chii-Min Hwu, Liang-Yu Lin.

**Formal analysis:** Tsung-Hui Wu, Chia-Huei Chu.

**Funding acquisition:** Li-Hsin Chang, Liang-Yu Lin.

**Investigation:** Liang-Yu Lin.

**Methodology:** Li-Hsin Chang, Chia-Huei Chu, Liang-Yu Lin.

**Resources:** Liang-Yu Lin.

**Supervision:** Harn-Shen Chen, Liang-Yu Lin.

**Writing – original draft:** Tsung-Hui Wu.

**Writing – review & editing:** Li-Hsin Chang, Chia-Huei Chu, Chii-Min Hwu, Harn-Shen Chen, Liang-Yu Lin.

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
