## [Decision Letter · Decision Letter 0]

31 Jan 2022

PONE-D-21-32495Soluble tumor necrosis factor receptor 2 are associated with progressive diabetic kidney disease in patients with type 2 diabetes mellitusPLOS ONE

Dear Dr. Lin,

Thank you for submitting your manuscript to PLOS ONE. After careful consideration, we feel that it has merit but does not fully meet PLOS ONE’s publication criteria as it currently stands. Therefore, we invite you to submit a revised version of the manuscript that addresses the points raised during the review process.

We look forward to receiving your revised manuscript.

Kind regards,

Karin Jandeleit-Dahm

Academic Editor

PLOS ONE

Journal Requirements:

 This study was supported by research grants V104E11-004-MY2, V105C-131, V107C-201, V108C-197, V109C-179 and V110C-198 provided to L.Y.L. from Taipei Veterans General Hospital and No. 2021001 to L.H.C. from Yeezen General Hospital.

This study was supported by research grants V104E11-004-MY2, V105C-131, V107C-201, V108C-197, V109C-179 and V110C-198 provided to L.Y.L. from Taipei Veterans General Hospital and No. 2021001 to L.H.C. from Yeezen General Hospital. 

 This study was supported by research grants V104E11-004-MY2, V105C-131, V107C-201, V108C-197, V109C-179 and V110C-198 provided to L.Y.L. from Taipei Veterans General Hospital and No. 2021001 to L.H.C. from Yeezen General Hospital.

Additional Editor Comments:

Please clarify any potential overlap with your previous publication in Oct 2021 (Endocrine Practice)- what is the novelty of this publication? What is the advantage of measuring soluble TNF receptor 2 over other biomarkers?

Reviewers' comments:

Reviewer's Responses to Questions

**Comments to the Author**

1. Is the manuscript technically sound, and do the data support the conclusions?

Reviewer #1: Partly

Reviewer #2: Yes

2. Has the statistical analysis been performed appropriately and rigorously? 

Reviewer #1: Yes

Reviewer #2: Yes

3. Have the authors made all data underlying the findings in their manuscript fully available?

Reviewer #1: No

Reviewer #2: Yes

4. Is the manuscript presented in an intelligible fashion and written in standard English?

Reviewer #1: Yes

Reviewer #2: Yes

5. Review Comments to the Author

Reviewer #1: Wu et al , investigated serum levels of TNFR2 in 364 type 2 diabetic patients to determine whether it was a valid biomarker for the onset of early diabetic nephropathy. The authors demonstrated a clear correlation between elevated levels of TNFR2 with decline in renal function ( increased albuminuria and reduced egfr). Whilst the data is presented well within the manuscript there are some concerns which the authors should address before this manuscript should be considered for publication:

1. The authors published data on 284 type 2 diabetic patients investigating TNFR1 as a biomarker and predictor of renal outcomes in Endocrinology practice , Oct 2021. The authors should verify and clarify that these patients are not indeed overlapping. Futhermore, the authors need to verify that the analysis was not conducted for both markers in the same cohort and published as two seperate manuscripts.

2. If they are indeed the same cohort or overlapping cohort the authors should provide a comparitive analysis in this new manuscript to determine which of the two markers are a a better and more valid biomarker of diabetic kidney disease.

3. The authors have presented many tables in this current manuscript with their hazard ratios and statistical analysis. I would suggest that the most significant asssociations, eGFR and albuminuria should be presented as a figure to further strenghten the findings of this manuscript.

4. Niewcaz et al published in 2012 that both TNFR1 and TNFR2 are both predictors of ESRD. The authors should clarify how their present study is adding to the previously published data. They also presented their findings in males and your cohort is primarily males. Is this a male phenomenon? You should compare males to females to answer this.

5. Figure 2A and 2B should be presented in higher quality images. It is not clear or of high enough standard for publication.

6. In table 3 when comparing the different models- Model 3 appears to have caused a loss in significance. The authors should address why the believe this to be the case and discuss it in their discussion.

Overall, I believe the authors should first and foremost verify that this current manuscript has not overlap with their publication in Oct 2021. If it does, clarify why is has been published in this way or at the very least make it clear. I think this should be the first priority to be addressed before publication is considered.

Reviewer #2: The manuscript under review investigates TNFR2 as a biomarker and a predictor of progressive renal disease in a cohort of diabetic patients. It investigates the relationship between serum TNFR2 levels and a number of parameters to conclude that serum TNFR2 levels above 1.608ng/ml are independently associated with a decline in eGFR and deteriorating albuminuria in T2D patients.

The only comments I have are:

1. The authors should include in their discussion other papers that have investigated this issue and how this work is different to theirs. For example there is no mention of PMID 22266663 and PMID 22266664, etc.

2. The authors should elaborate how TNFR2 as a predictor of progressive kidney disease compares to other markers that are currently used. Is it an improvement? Is it an easier marker to measure?

6. PLOS authors have the option to publish the peer review history of their article (what does this mean?). If published, this will include your full peer review and any attached files.

Reviewer #1: No

Reviewer #2: No

---

## [Author Response · Author response to Decision Letter 0]

3 Mar 2022

Academic Editor

Answer to academic editor:

 Thanks for your kindly remind. We revise our manuscript in adherence to PLOS ONE's submission guidelines. We resubmit all required files for revision and hope that the changes made will be considered satisfactory.

Answer to academic editor:

 Thanks for your suggestion. The participants included in our study are all adults. Written informed consent was obtained from each participant before entering the study. We revise the manuscript as below:

Line 105, page 8, part of “Study design and participants”, section of “Materials and methods”

Original

This prospective study was approved by the ethics committee at Taipei Veterans General Hospital. Informed consent was obtained from all participants, and the study complied with the guidelines of the Declaration of Helsinki.

Revised

This prospective study was approved by the ethics committee at Taipei Veterans General Hospital. Written informed consent was obtained from each participant before entering the study. The study complied with the guidelines of the Declaration of Helsinki.

This study was supported by research grants V104E11-004-MY2, V105C-131, V107C-201, V108C-197, V109C-179 and V110C-198 provided to L.Y.L. from Taipei Veterans General Hospital and No. 2021001 to L.H.C. from Yeezen General Hospital.

Answer to academic editor:

 Thanks for your suggestion. The funders in our study had no role in study design, data collection and analysis, decision to publish, or preparation of the manuscript. We add this information in the cover letter.

Cover letter

Original

This manuscript has not been published or presented elsewhere in part or in entirety, and is not under consideration by another journal. All study participants provided informed consent, and the study design was approved by the appropriate ethics review boards. All the authors have approved the manuscript and agree with submission to your esteemed journal. There are no conflicts of interest to declare.

Revised

This manuscript has not been published or presented elsewhere in part or in entirety, and is not under consideration by another journal. All study participants provided informed consent, and the study design was approved by the appropriate ethics review boards. All the authors have approved the manuscript and agree with submission to your esteemed journal. The funders had no role in study design, data collection and analysis, decision to publish, or preparation of the manuscript. There are no conflicts of interest to declare.

This study was supported by research grants V104E11-004-MY2, V105C-131, V107C-201, V108C-197, V109C-179 and V110C-198 provided to L.Y.L. from Taipei Veterans General Hospital and No. 2021001 to L.H.C. from Yeezen General Hospital.

Please note that funding information should not appear in the Acknowledgments section or other areas of your manuscript. We will only publish funding information present in the Funding Statement section of the online submission form. Please remove any funding-related text from the manuscript and let us know how you would like to update your Funding Statement. Currently, your Funding Statement reads as follows:

This study was supported by research grants V104E11-004-MY2, V105C-131, V107C-201, V108C-197, V109C-179 and V110C-198 provided to L.Y.L. from Taipei Veterans General Hospital and No. 2021001 to L.H.C. from Yeezen General Hospital.

Answer to academic editor:

 Thanks for your kindly remind. Besides previous funding statement, this study was supported by research grants V111D62-002-MY3-1 from Taipei Veterans General Hospital as well. We add this information within our cover letter. The funding-related text was removed from the manuscript.

Line 340, page 24, section of “Acknowledgments”

Original

This study was supported by research grants V104E11-004-MY2, V105C-131, V107C-201, V108C-197, V109C-179 and V110C-198 provided to L.Y.L. from Taipei Veterans General Hospital and No. 2021001 to L.H.C. from Yeezen General Hospital. The authors thank the Medical Sciences & Technology Building of Taipei Veterans General Hospital for providing experimental space and facilities. The authors declare no conflicts of interest regarding the contents of this manuscript.

Revised

The authors thank the Medical Sciences & Technology Building of Taipei Veterans General Hospital for providing experimental space and facilities. The authors declare no conflicts of interest regarding the contents of this manuscript.

Cover letter

Original

This manuscript has not been published or presented elsewhere in part or in entirety, and is not under consideration by another journal. All study participants provided informed consent, and the study design was approved by the appropriate ethics review boards. All the authors have approved the manuscript and agree with submission to your esteemed journal. The funders had no role in study design, data collection and analysis, decision to publish, or preparation of the manuscript. There are no conflicts of interest to declare.

Revised

This manuscript has not been published or presented elsewhere in part or in entirety, and is not under consideration by another journal. All study participants provided informed consent, and the study design was approved by the appropriate ethics review boards. All the authors have approved the manuscript and agree with submission to your esteemed journal. This study was supported by research grants V104E11-004-MY2, V105C-131, V107C-201, V108C-197, V109C-179, V110C-198, and V111D62-002-MY3-1 provided to L.Y.L. from Taipei Veterans General Hospital and No. 2021001 to L.H.C. from Yeezen General Hospital. The funders had no role in study design, data collection and analysis, decision to publish, or preparation of the manuscript. There are no conflicts of interest to declare.

5. Please clarify any potential overlap with your previous publication in Oct 2021 (Endocrine Practice)- what is the novelty of this publication? What is the advantage of measuring soluble TNF receptor 2 over other biomarkers?

Answer to academic editor:

 Thanks for your suggestion. In our prospective observational cohort study, we started to enroll patients with type 2 diabetes mellitus who visited the endocrinology and metabolism outpatient clinic at Taipei Veterans General Hospital since June 2014. We measured serum sTNFR1 levels initially while 283 participants were enrolled in the study. Our investigation suggested that increased sTNFR1 levels were associated with renal outcomes in subjects with type 2 diabetes mellitus, which was published in Endocrine Practice (doi: 10.4158/EP-2020-0114, PMID: 32576039). Subsequently, we measured serum sTNFR2 concentrations when 364 patients were enrolled. The participants were overlapping but more patients were recruited comparing to the previous study. Our analysis also proved that circulating levels of sTNFR2 are associated with the progression of diabetic kidney disease in patients with type 2 diabetes independent of baseline albuminuria or renal function. This cohort study is still ongoing, and more patients will be enrolled.

 Both serum sTNFR1 and sTNFR2 levels were measured in the 364 patients enrolled in the current study. The mean areas under the ROC curve were 0.64 (95% CI 0.57-0.71, p < 0.001) for sTNFR1 and 0.63 (95% CI 0.57-0.70, p < 0.001) for sTNFR2. The difference between areas under the ROC curve of sTNFR1 and sTNFR2 is not significant (p = 0.952). Both sTNFR1 and sTNFR2 are potential biomarkers in the assessment of diabetic kidney disease.

 The clinical methods for screening diabetic nephropathy includes annual measurement of urinary albumin-creatinine ratio (UACR) and estimated glomerular filtration rate (eGFR). However, significant glomerular structural changes often existed when microalbuminuria becomes apparent. Patients with type 2 diabetes can also develop nonproteinuric diabetic nephropathy, and measuring UACR is an ineffective screening test for these patients. In our study, circulating sTNFR2 levels are associated with worsening albuminuria and progressive eGFR decline in patients with type 2 diabetes, including participants with eGFRs ≥60 mL/min/1.73m2 or normoalbuminuria. Our findings suggested that sTNFR2 is a biomarker to detect earlier stages of diabetic kidney disease comparing to measurement of UACR and eGFR. Measuring serum levels of sTNFR2 is a noninvasive and easily accessible method, making it a potential screening tool for diabetic nephropathy. We revise the manuscript as below:

Line 247, page 19, section of “Discussion”

Original

In our study, elevated serum sTNFR2 concentrations were associated with worsening albuminuria and progressive eGFR decline in patients with type 2 diabetes. Circulating sTNFR2 levels ≥1.608 ng/mL were a strong predictor of the progression of albuminuria and renal impairment in patients with type 2 diabetes, and the association was independent of relevant clinical covariates, including age, duration of diabetes, baseline UACR and eGFR, and the use of RAS inhibitors.

Revised

In our study, elevated serum sTNFR2 concentrations were associated with worsening albuminuria and progressive eGFR decline in patients with type 2 diabetes. Circulating sTNFR2 levels ≥1.608 ng/mL were a strong predictor of the progression of albuminuria and renal impairment in patients with type 2 diabetes, and the association was independent of relevant clinical covariates, including age, duration of diabetes, baseline UACR and eGFR, and the use of RAS inhibitors. Comparing to the current screening strategies of diabetic nephropathy as measurement of UACR and eGFR, sTNFR2 is a biomarker to detect earlier stages of diabetic kidney disease. As a noninvasive and easily accessible method, measuring serum levels of sTNFR2 is a potential screening tool for diabetic nephropathy.

Line 285, page 21, section of “Discussion”

Original

We previously reported that serum sTNFR1 levels were associated with renal outcomes in patients with type 2 diabetes [9]. The association of serum sTNFR1 levels above the cut-off value yielded by the receiver operating characteristic curve analysis with DKD progression was significant after adjusting for relevant clinical variables (HR 2.43, 95% CI 1.18-5.02, p = 0.01). Although the hazard ratios calculated for renal outcomes according to the serum sTNFR1 levels were higher than those calculated for serum sTNFR2 levels, the CIs for the hazard ratios overlapped.

Revised

We previously reported that serum sTNFR1 levels were associated with renal outcomes in patients with type 2 diabetes while 283 participants were enrolled in the same cohort in this study [9]. The association of serum sTNFR1 levels above the cut-off value yielded by the receiver operating characteristic curve analysis with DKD progression was significant after adjusting for relevant clinical variables (HR 2.43, 95% CI 1.18-5.02, p = 0.01). Although the hazard ratios calculated for renal outcomes according to the serum sTNFR1 levels were higher than those calculated for serum sTNFR2 levels, the CIs for the hazard ratios overlapped. In addition, both serum sTNFR1 and sTNFR2 levels were measured in the 364 patients enrolled in the current study. The difference between areas under the ROC curve of sTNFR1 and sTNFR2 is not significant (data not shown). Both sTNFR1 and sTNFR2 are potential biomarkers in the assessment of diabetic kidney disease.

 

Reviewer 1

Wu et al, investigated serum levels of TNFR2 in 364 type 2 diabetic patients to determine whether it was a valid biomarker for the onset of early diabetic nephropathy. The authors demonstrated a clear correlation between elevated levels of TNFR2 with decline in renal function (increased albuminuria and reduced egfr). Whilst the data is presented well within the manuscript there are some concerns which the authors should address before this manuscript should be considered for publication:

1. The authors published data on 284 type 2 diabetic patients investigating TNFR1 as a biomarker and predictor of renal outcomes in Endocrinology practice , Oct 2021. The authors should verify and clarify that these patients are not indeed overlapping. Futhermore, the authors need to verify that the analysis was not conducted for both markers in the same cohort and published as two seperate manuscripts.

Answer to the reviewer:

 Thanks for your comments. In our prospective observational cohort study, we started to enroll patients with type 2 diabetes mellitus who visited the endocrinology and metabolism outpatient clinic at Taipei Veterans General Hospital since June 2014. We measured serum sTNFR1 levels initially while 283 participants were enrolled in the study. Our investigation suggested that increased sTNFR1 levels were associated with renal outcomes in subjects with type 2 diabetes mellitus, which was published in Endocrine Practice (doi: 10.4158/EP-2020-0114, PMID: 32576039). Subsequently, we measured serum sTNFR2 concentrations when 364 patients were enrolled. The participants were overlapping but more patients were recruited comparing to the previous study. Our analysis also proved that circulating levels of sTNFR2 are associated with the progression of diabetic kidney disease in patients with type 2 diabetes independent of baseline albuminuria or renal function. This cohort study is still ongoing, and more patients will be enrolled. We add this information in the manuscript:

Line 285, page 21, section of “Discussion”

Original

We previously reported that serum sTNFR1 levels were associated with renal outcomes in patients with type 2 diabetes [9].

Revised

We previously reported that serum sTNFR1 levels were associated with renal outcomes in patients with type 2 diabetes while 283 participants were enrolled in the same cohort in this study [9].

2. If they are indeed the same cohort or overlapping cohort the authors should provide a comparitive analysis in this new manuscript to determine which of the two markers are a a better and more valid biomarker of diabetic kidney disease.

Answer to the reviewer:

 Thanks for your suggestion. Both serum sTNFR1 and sTNFR2 levels were measured in the 364 patients enrolled in the current study. The mean areas under the ROC curve were 0.64 (95% CI 0.57-0.71, p < 0.001) for sTNFR1 and 0.63 (95% CI 0.57-0.70, p < 0.001) for sTNFR2. The difference between areas under the ROC curve of sTNFR1 and sTNFR2 is not significant (p = 0.952). Both sTNFR1 and sTNFR2 are potential biomarkers in the assessment of diabetic kidney disease.

 We add this information in the manuscript:

Line 291, page 21, section of “Discussion”

Original

We previously reported that serum sTNFR1 levels were associated with renal outcomes in patients with type 2 diabetes while 283 participants were enrolled in the same cohort in this study [9]. The association of serum sTNFR1 levels above the cut-off value yielded by the receiver operating characteristic curve analysis with DKD progression was significant after adjusting for relevant clinical variables (HR 2.43, 95% CI 1.18-5.02, p = 0.01). Although the hazard ratios calculated for renal outcomes according to the serum sTNFR1 levels were higher than those calculated for serum sTNFR2 levels, the CIs for the hazard ratios overlapped.

Revised

We previously reported that serum sTNFR1 levels were associated with renal outcomes in patients with type 2 diabetes while 283 participants were enrolled in the same cohort in this study [9]. The association of serum sTNFR1 levels above the cut-off value yielded by the receiver operating characteristic curve analysis with DKD progression was significant after adjusting for relevant clinical variables (HR 2.43, 95% CI 1.18-5.02, p = 0.01). Although the hazard ratios calculated for renal outcomes according to the serum sTNFR1 levels were higher than those calculated for serum sTNFR2 levels, the CIs for the hazard ratios overlapped. In addition, both serum sTNFR1 and sTNFR2 levels were measured in the 364 patients enrolled in the current study. The difference between areas under the ROC curve of sTNFR1 and sTNFR2 is not significant (data not shown). Both sTNFR1 and sTNFR2 are potential biomarkers in the assessment of diabetic kidney disease.

3. The authors have presented many tables in this current manuscript with their hazard ratios and statistical analysis. I would suggest that the most significant asssociations, eGFR and albuminuria should be presented as a figure to further strenghten the findings of this manuscript.

Answer to the reviewer:

 Thanks for your suggestion. To highlight our findings, we plot cumulative hazard curves for renal composite events in patients with type 2 diabetes based on the threshold value of soluble tumor necrosis factor receptor 2 (≥1.608 ng/mL vs. <1.608 ng/mL) in model 3. Analysis results are adjusted for age, sex, duration of diabetes, retinopathy, neuropathy, systolic blood pressure, hemoglobin A1C, estimated glomerular filtration rate, urinary albumin-creatinine ratio, renin-angiotensin system inhibitors, and diuretics. We revise the manuscript as below:

Line 216, page 16, part of “Association of sTNFR2 levels ≥1.608 ng/mL with the renal outcomes”, section of “Results”

Original

In the univariate Cox proportional analysis, sTNFR2 concentrations ≥1.608 ng/mL were associated with higher renal event rates (HR 2.47, 95% CI 1.45-4.21, p = 0.001). A sTNFR2 concentration ≥1.608 ng/mL predicted renal events in all of the multivariate models (HR 2.41, 95% CI 1.38-4.21, p = 0.002 for model 1; HR 2.31, 95% CI 1.32-4.04, p = 0.003 for model 2; HR 2.27, 95% CI 1.23-4.20, p = 0.009 for model 3).

Revised

In the univariate Cox proportional analysis, sTNFR2 concentrations ≥1.608 ng/mL were associated with higher renal event rates (HR 2.47, 95% CI 1.45-4.21, p = 0.001). A sTNFR2 concentration ≥1.608 ng/mL predicted renal events in all of the multivariate models (HR 2.41, 95% CI 1.38-4.21, p = 0.002 for model 1; HR 2.31, 95% CI 1.32-4.04, p = 0.003 for model 2; HR 2.27, 95% CI 1.23-4.20, p = 0.009 for model 3; Fig 3).

Fig 3.

Cumulative hazard curves for renal events (a composite of a >30% reduction of the estimated glomerular filtration rate and/or the progression of albuminuria stages) in patients with type 2 diabetes based on the threshold value of soluble tumor necrosis factor receptor 2 (≥1.608 ng/mL vs. <1.608 ng/mL). Analysis results were adjusted for age, sex, duration of diabetes, retinopathy, neuropathy, systolic blood pressure, hemoglobin A1C, estimated glomerular filtration rate, urinary albumin-creatinine ratio, renin-angiotensin system inhibitors, and diuretics.

4. Niewcaz et al published in 2012 that both TNFR1 and TNFR2 are both predictors of ESRD. The authors should clarify how their present study is adding to the previously published data. They also presented their findings in males and your cohort is primarily males. Is this a male phenomenon? You should compare males to females to answer this.

Answer to the reviewer:

 Thanks for your suggestion. Niewczas et al. demonstrated that higher serum sTNFR2 concentrations are associated with increased risks of end stage renal disease in patients with type 2 diabetes. We have cited the article as reference 11 in the second paragraph of Discussion. In our study, the renal outcomes are defined as a composite of a >30% reduction of the estimated glomerular filtration rate and/or the progression of albuminuria stages. Comparing to the previous study mainly focusing on advanced diabetic kidney disease, the findings in our study suggested that sTNFR2 is a promising predictor of the early stages of diabetic nephropathy.

 In our study, participants were enrolled from a veterans hospital which mainly serves male patients. It accounts for male-predominant cohort in this study. In the female participants, the association between sTNFR2 concentration ≥1.608 ng/mL and renal composite events was significant initially but became attenuated after adjustment in model 2 and model 3 (Crude HR 3.35, 95% CI 1.15-9.77, p = 0.027; HR 3.53, 95% CI 1.19-10.49, p = 0.023 for model 1; HR 3.00, 95% CI 0.98-9.15, p = 0.053 for model 2; HR 4.12, 95% CI 1.00-17.07, p = 0.051 for model 3). The borderline results in model 2 (p = 0.053) and model 3 (p = 0.051) may be due to relatively small sample size. Further studies are needed to confirm the association between sTNFR2 and renal outcomes among female patients with type 2 diabetes.

Model 1: adjusted for age and sex.

Model 2: adjusted for age, sex, and duration of diabetes.

Model 3: adjusted for age, sex, duration of diabetes, retinopathy, neuropathy, systolic blood pressure, HbA1C, eGFR, UACR, RAS inhibitors, and diuretics.

 We revise the manuscript as below:

Line 270, page 20, section of “Discussion”

Original

Higher serum sTNFR2 concentrations have been proven to be associated with increased risks of renal function decline or ESRD in patients with type 2 diabetes [10-13]. In one cohort of American Indians (Pima Indians) with type 2 diabetes, patients with macroalbuminuria whose serum sTNFR2 levels were in the highest quartile had a 88.7% cumulative incidence of ESRD, whereas those with macroalbuminuria and sTNFR2 levels in the lowest three quartiles had a 47.3% cumulative incidence of ESRD at 10 years of follow-up [12]. Higher sTNFR2 levels were associated with all-cause mortality as well [10]. The findings in our cohort suggested that sTNFR2 is a promising biomarker in the early stages of diabetic nephropathy. The association of sTNFR2 concentration ≥1.608 ng/mL with renal outcomes remained significant in all subgroups, including patients with eGFRs ≥60 mL/min/1.73m2 or normoalbuminuria.

Revised

Higher serum sTNFR2 concentrations have been proven to be associated with increased risks of renal function decline or ESRD in patients with type 2 diabetes [10-13]. In one cohort of American Indians (Pima Indians) with type 2 diabetes, patients with macroalbuminuria whose serum sTNFR2 levels were in the highest quartile had a 88.7% cumulative incidence of ESRD, whereas those with macroalbuminuria and sTNFR2 levels in the lowest three quartiles had a 47.3% cumulative incidence of ESRD at 10 years of follow-up [12]. Higher sTNFR2 levels were associated with all-cause mortality as well [10]. Comparing to previous studies mainly focusing on advanced diabetic kidney disease, the findings in our cohort suggested that sTNFR2 is a promising biomarker in the early stages of diabetic nephropathy. The association of sTNFR2 concentration ≥1.608 ng/mL with renal outcomes remained significant in all subgroups, including patients with eGFRs ≥60 mL/min/1.73m2 or normoalbuminuria.

Line 228, page 17, part of “Association of sTNFR2 levels ≥1.608 ng/mL with the renal outcomes”, section of “Results”

Original

We performed a subgroup analysis based on age (younger or older than 60 years), sex, systolic blood pressure (≥ or <140 mmHg), eGFR (≥ or <60 mL/min/1.73m2), UACR (≥ or <30 mg/g creatinine), and the use of RAS inhibitors. The association of a sTNFR2 concentration ≥1.608 ng/mL with renal outcomes remained significant across all subgroups, including patients with an eGFR ≥60 mL/min/1.73m2 or normoalbuminuria (Table 3).

Revised

We performed a subgroup analysis based on age (younger or older than 60 years), sex, systolic blood pressure (≥ or <140 mmHg), eGFR (≥ or <60 mL/min/1.73m2), UACR (≥ or <30 mg/g creatinine), and the use of RAS inhibitors. The association of a sTNFR2 concentration ≥1.608 ng/mL with renal outcomes remained significant across all subgroups, including patients with an eGFR ≥60 mL/min/1.73m2 or normoalbuminuria (Table 3). In the female participants, the association between sTNFR2 concentration ≥1.608 ng/mL and renal composite events was significant initially but became attenuated after adjustment in model 2 and model 3 (Crude HR 3.35, 95% CI 1.15-9.77, p = 0.027; HR 3.53, 95% CI 1.19-10.49, p = 0.023 for model 1; HR 3.00, 95% CI 0.98-9.15, p = 0.053 for model 2; HR 4.12, 95% CI 1.00-17.07, p = 0.051 for model 3).

Line 320, page 23, section of “Discussion”

Original

The limitations of our study should be considered. First, the participants in this cohort were recruited from a single center, and we cannot avoid selection bias completely.

Revised

The limitations of our study should be considered. First, the participants in this cohort were recruited from a veterans hospital which mainly serves male patients. It accounts for male-predominant cohort in this study. In the female participants, the borderline results of the association between sTNFR2 concentration ≥1.608 ng/mL and renal composite events may be due to relatively small sample size.

5. Figure 2A and 2B should be presented in higher quality images. It is not clear or of high enough standard for publication.

Answer to the reviewer:

 Thanks for your kindly remind. We follow the submission guidelines of PLOS ONE and upload our figure files to the Preflight Analysis and Conversion Engine (PACE). All of the figures are adjusted to meet requirements of PLOS ONE.

6. In table 3 when comparing the different models- Model 3 appears to have caused a loss in significance. The authors should address why the believe this to be the case and discuss it in their discussion.

Answer to the reviewer:

 Thanks for your comments. Chronic low-grade inflammation plays an important role in the development of diabetic kidney disease. The tumor necrosis factor-α pathway, including sTNFR2, has emerged as an important mechanism in the pathogenesis of diabetic kidney disease. Our finding suggested that circulating sTNFR2 levels ≥1.608 ng/mL were a strong predictor of the progression of albuminuria and renal impairment in patients with type 2 diabetes. However, the significance of the association between serum sTNFR2 levels and the renal outcomes disappear after adjusting several confounding factors in model 3. The failure to detect significant association may be due to inadequate sample size and duration of follow-up. Further research is needed to confirm the association of sTNFR2 with renal outcomes in diabetic patients. We revise the manuscript as below:

Line 275, page 20, section of “Discussion”

Original

The findings in our cohort suggested that sTNFR2 is a promising biomarker in the early stages of diabetic nephropathy. The association of sTNFR2 concentration ≥1.608 ng/mL with renal outcomes remained significant in all subgroups, including patients with eGFRs ≥60 mL/min/1.73m2 or normoalbuminuria.

Revised

The findings in our cohort suggested that sTNFR2 is a promising biomarker in the early stages of diabetic nephropathy. The association of sTNFR2 concentration ≥1.608 ng/mL with renal outcomes remained significant in all subgroups, including patients with eGFRs ≥60 mL/min/1.73m2 or normoalbuminuria. 

The significance of the association between serum sTNFR2 levels and the renal outcomes disappear after adjusting several confounding factors in model 3 (Table 2). The failure to detect significant association may be due to inadequate sample size and duration of follow-up. Further research is needed to confirm the association of sTNFR2 with renal outcomes in diabetic patients.

 

Reviewer 2

 The manuscript under review investigates TNFR2 as a biomarker and a predictor of progressive renal disease in a cohort of diabetic patients. It investigates the relationship between serum TNFR2 levels and a number of parameters to conclude that serum TNFR2 levels above 1.608ng/ml are independently associated with a decline in eGFR and deteriorating albuminuria in T2D patients.

 The only comments I have are:

1. The authors should include in their discussion other papers that have investigated this issue and how this work is different to theirs. For example there is no mention of PMID 22266663 and PMID 22266664, etc.

Answer to the reviewer:

 Thanks for your suggestion. Niewczas et al. demonstrated that higher serum sTNFR2 concentrations are associated with increased risks of end stage renal disease in patients with type 2 diabetes. We have cited the article as reference 11 in the second paragraph of Discussion. Gohda et al. proved that circulating sTNFR2 predicts stage 3 chronic kidney disease in patients with type 1 diabetes. In our study, the renal outcomes are defined as a composite of a >30% reduction of the estimated glomerular filtration rate and/or the progression of albuminuria stages. Comparing to previous studies mainly focusing on advanced diabetic kidney disease, the findings in our study suggested that sTNFR2 is a promising predictor of the early stages of diabetic nephropathy. We revise the manuscript as below:

Line 264, page 20, section of “Discussion”

Original

Higher serum sTNFR2 concentrations have been proven to be associated with increased risks of renal function decline or ESRD in patients with type 2 diabetes [10-13]. In one cohort of American Indians (Pima Indians) with type 2 diabetes, patients with macroalbuminuria whose serum sTNFR2 levels were in the highest quartile had a 88.7% cumulative incidence of ESRD, whereas those with macroalbuminuria and sTNFR2 levels in the lowest three quartiles had a 47.3% cumulative incidence of ESRD at 10 years of follow-up [12]. Higher sTNFR2 levels were associated with all-cause mortality as well [10]. The findings in our cohort suggested that sTNFR2 is a promising biomarker in the early stages of diabetic nephropathy. The association of sTNFR2 concentration ≥1.608 ng/mL with renal outcomes remained significant in all subgroups, including patients with eGFRs ≥60 mL/min/1.73m2 or normoalbuminuria.

Revised

Higher serum sTNFR2 concentrations have been proven to be associated with increased risks of renal function decline or ESRD in patients with type 2 diabetes [10-13] and chronic kidney disease in patients with type 1 diabetes [20]. In one cohort of American Indians (Pima Indians) with type 2 diabetes, patients with macroalbuminuria whose serum sTNFR2 levels were in the highest quartile had a 88.7% cumulative incidence of ESRD, whereas those with macroalbuminuria and sTNFR2 levels in the lowest three quartiles had a 47.3% cumulative incidence of ESRD at 10 years of follow-up [12]. Higher sTNFR2 levels were associated with all-cause mortality as well [10]. Comparing to previous studies mainly focusing on advanced diabetic kidney disease, the findings in our cohort suggested that sTNFR2 is a promising biomarker in the early stages of diabetic nephropathy. The association of sTNFR2 concentration ≥1.608 ng/mL with renal outcomes remained significant in all subgroups, including patients with eGFRs ≥60 mL/min/1.73m2 or normoalbuminuria.

2. The authors should elaborate how TNFR2 as a predictor of progressive kidney disease compares to other markers that are currently used. Is it an improvement? Is it an easier marker to measure?

Answer to the reviewer:

 Thanks for your comments. The clinical methods for screening diabetic nephropathy includes annual measurement of urinary albumin-creatinine ratio (UACR) and estimated glomerular filtration rate (eGFR). However, significant glomerular structural changes often existed when microalbuminuria becomes apparent. Patients with type 2 diabetes can also develop nonproteinuric diabetic nephropathy, and measuring UACR is an ineffective screening test for these patients. In our study, circulating sTNFR2 levels are associated with worsening albuminuria and progressive eGFR decline in patients with type 2 diabetes, including participants with eGFRs ≥60 mL/min/1.73m2 or normoalbuminuria. Our findings suggested that sTNFR2 is a biomarker to detect earlier stages of diabetic kidney disease comparing to measurement of UACR and eGFR. Measuring serum levels of sTNFR2 is a noninvasive and easily accessible method, making it a potential screening tool for diabetic nephropathy. We revise the manuscript as below:

Line 247, page 19, section of “Discussion”

Original

In our study, elevated serum sTNFR2 concentrations were associated with worsening albuminuria and progressive eGFR decline in patients with type 2 diabetes. Circulating sTNFR2 levels ≥1.608 ng/mL were a strong predictor of the progression of albuminuria and renal impairment in patients with type 2 diabetes, and the association was independent of relevant clinical covariates, including age, duration of diabetes, baseline UACR and eGFR, and the use of RAS inhibitors.

Revised

In our study, elevated serum sTNFR2 concentrations were associated with worsening albuminuria and progressive eGFR decline in patients with type 2 diabetes. Circulating sTNFR2 levels ≥1.608 ng/mL were a strong predictor of the progression of albuminuria and renal impairment in patients with type 2 diabetes, and the association was independent of relevant clinical covariates, including age, duration of diabetes, baseline UACR and eGFR, and the use of RAS inhibitors. Comparing to the current screening strategies of diabetic nephropathy as measurement of UACR and eGFR, sTNFR2 is a biomarker to detect earlier stages of diabetic kidney disease. As a noninvasive and easily accessible method, measuring serum levels of sTNFR2 is a potential screening tool for diabetic nephropathy.

---

## [Decision Letter · Decision Letter 1]

29 Mar 2022

Soluble tumor necrosis factor receptor 2 are associated with progressive diabetic kidney disease in patients with type 2 diabetes mellitus

PONE-D-21-32495R1

Dear Dr. Lin,

We’re pleased to inform you that your manuscript has been judged scientifically suitable for publication and will be formally accepted for publication once it meets all outstanding technical requirements.

Kind regards,

Karin Jandeleit-Dahm

Academic Editor

PLOS ONE

Additional Editor Comments (optional):

Thank you for responding to the queries. Please amend title: Soluble tumour necrosis factor receptor 2 "is"

Reviewers' comments:

Reviewer's Responses to Questions

**Comments to the Author**

1. If the authors have adequately addressed your comments raised in a previous round of review and you feel that this manuscript is now acceptable for publication, you may indicate that here to bypass the “Comments to the Author” section, enter your conflict of interest statement in the “Confidential to Editor” section, and submit your "Accept" recommendation.

Reviewer #1: (No Response)

Reviewer #2: All comments have been addressed

2. Is the manuscript technically sound, and do the data support the conclusions?

Reviewer #1: Partly

Reviewer #2: Yes

3. Has the statistical analysis been performed appropriately and rigorously? 

Reviewer #1: Yes

Reviewer #2: Yes

4. Have the authors made all data underlying the findings in their manuscript fully available?

Reviewer #1: Yes

Reviewer #2: Yes

5. Is the manuscript presented in an intelligible fashion and written in standard English?

Reviewer #1: Yes

Reviewer #2: Yes

6. Review Comments to the Author

Reviewer #1: Whist the authors have clarified that 284 of these patients used in the current study have been used for an earlier publication. I do not believe that this current manuscript can be considered as a novel publication.

Reviewer #2: No further comments. All my issues have been addressed. The revised manuscript is considerably improved.

7. PLOS authors have the option to publish the peer review history of their article (what does this mean?). If published, this will include your full peer review and any attached files.

Reviewer #1: No

Reviewer #2: No

---

## [Editor Report · Acceptance letter]

1 Apr 2022

PONE-D-21-32495R1 

Soluble tumor necrosis factor receptor 2 is associated with progressive diabetic kidney disease in patients with type 2 diabetes mellitus 

Dear Dr. Lin:

I'm pleased to inform you that your manuscript has been deemed suitable for publication in PLOS ONE. Congratulations! Your manuscript is now with our production department. 

Kind regards, 

on behalf of

Professor Karin Jandeleit-Dahm 

Academic Editor

PLOS ONE